# The Effect of Soil Surface Mounds and Depressions on Runoff

Xinlan Liang [1], Jiawei Feng [1], Zhixin Ye [1], Lei Zhang [1], Jidong Li [1], Xiuyuan Lu [1], Sixiang Zhao [1], Qi Liu [1], Zicheng Zheng [2] and Yong Wang [3,*]

[1] College of Water Conservancy and Hydropower Engineering, Sichuan Agricultural University, Ya'an 625014, China
[2] College of Resources, Sichuan Agricultural University, Chengdu 611130, China
[3] College of Forestry, Sichuan Agricultural University, Chengdu 611130, China
* Correspondence: wangyong2015@sicau.edu.cn; Tel.: +86-186-8344-1913

**Abstract:** Surface mounds and depressions are the basic patterns of microtopography. Their geometric forms and physical properties affect rainfall infiltration, runoff generation and runoff confluence process. In this study, soil beds were set up with seven different types of microtopography to study the effects of surface mounds and depressions on runoff. They were the control check (CK), alternate mounds (AM), continuous mounds (CM), alternate depressions (AD), continuous depressions (CD), alternate mounds and depressions (AMD) and continuous mounds and depressions (CMD). There was only one microtopography type for monomorphic surface relief (MSR) while two for compound surface relief (CSR). All soil beds were exposed under 60, 90 or 120 mm/h rainfall intensity for 90 min. The main results are as follows: surface mounds could promote surface runoff, triggering and shortening runoff generation time, while surface depressions showed contrary results. Whether there was an interval between mounds or depressions also affected the characteristics of runoff. The runoff generation time was 3.8–5.0 times higher for continuous slope than for interval slope, while the runoff yield and runoff coefficient both decreased by approximately 40%. CSR can significantly neutralize the flow-promoting effects of the mounds and the flow-inhibiting effects of the depressions, making the runoff yield and runoff process present a neutral state between the mounds and depressions. CSR prolongs runoff generation time from 1–10 min of MSR to 5–16 min. The runoff yield of CSR presented as 0.12, between 0.17 for mounds and 0.10 for depressions, and so did the runoff coefficient and hydrodynamic parameters. In addition, with rainfall intensity increased, the runoff pattern of CSR and MSR became more similar to each other, and the retarding effects of topography on overland flow were more effective.

**Keywords:** microrelief; interaction; monomorphic surface relief; compound surface relief; hydrodynamic characteristics

## 1. Introduction

As the most common form of erosion, water erosion is a complicated process with spatial and temporal variability, and because of the dynamics of runoff [1], it causes serious long-term damages to surface soil in the whole world [2,3]. There are several factors that influence slope runoff, such as soil properties, rainfall intensity, rainfall patterns, the drop size and kinetic energy [1,4,5], microtopography [6,7] as well as tillage and management practices [8,9]. Microtopography, also called soil surface microrelief, is the slight change of surface elevation caused by tillage method and soil erosion; it affects many aspects of runoff, such as depression storage, infiltration, runoff velocity and organization [10,11].

Because of the raised and sunken microtopography, soil surface roughness changes the water storage and alters the flow direction, and ultimately affects the whole runoff process [12]. The physical structure characteristics of surface mounds and depressions play different roles in runoff generation, confluence, infiltration, filling and storage [13,14]. For example, Darboux et al. (2002) examined the role of surface mounds and depressions

on overland flow [12]. They observed that the distribution of overland flow was not uniform due to the local heterogeneity of the flow formed by the mounds and depressions. Obviously, due to the shape of the mounds and the depressions, the depth of the slope flow around the convex section is shallower, while the depth of the slope flow around the depressions is deeper. The mounds showed stronger permeability than plain surface through an ink infiltration test [15], while the storage volume and spatial distributions of depressions dominated their impact capacities [16]. It can be seen that the surface mounds and depression showed different effects on runoff process.

Many relevant studies have been done on the mechanism of runoff generation under different surface types [17] and different tillage practices [18,19]. What is more, the influence of different surface microrelief on the runoff yield and confluence mechanism has also been studied [20–22]. Previous studies on the relationship between microrelief and runoff showed basically similar results, that is, surface microrelief would increase infiltration and decrease runoff [14,23]. However, these results were based on the coexistence of both mounds and depressions, including the interaction of them. The separate role of mounds and depressions were not proved yet.

In this study, soil surface mounds were set up to form the raised microtopography while depressions were used to form sunken microtopography. As the differences in runoff generation are limited among plots of different slope gradients [24], only 9% slope gradient was chosen, since the main objective of this study was to reveal the mechanism of soil surface mounds and depressions on runoff generation and runoff processing, as well as the potential interaction of mounds and depressions.

This research was conducted in order to reveal the separate role of mounds and depressions on runoff, and to pave the way for the subsequent study of the separate effect of mounds and depressions on water erosion. In view of this, the main task of this research was to separately study how the raised and sunken microtopography affect the runoff generation time, runoff yield, runoff processing and the hydrodynamic characteristics. Through the ways of simulated rainfall and close-range photogrammetry, the surface mounds and depressions were studied individually and conjunctively to clarify the different mechanisms of these two topographies of runoff triggering and process, in order to improve further study for the effect of microtopography on soil erosion.

## 2. Materials and Methods

### 2.1. Site Description

The experiment was conducted in the Scientific Research Center of Sichuan Agricultural University in Yucheng District, Ya'an City, Sichuan Province, China, from May to December 2019. This center is located in the mountainous area of the western edge of Sichuan Basin. It is the transition zone from the Tibet Plateau to the basin. The terrain is high in the north, west and south, while low in the middle and east [25]. The climate type is subtropical monsoon humid; the annual average temperature is more than 14 °C, the abundant rainfall here makes it so the annual rainy days are more than 200 days and the average annual precipitation is about 1800 mm. This region having the lowest difference in temperature and the highest precipitation make Ya'an unique compared to other subtropical monsoon cities in the same latitude region in China.

The soil type was purple soil, which is a common soil type in Sichuan Basin. A total of 5 sample sites were selected in the test fields, and soil samples were collected and sent back to the laboratory to measure the physical and chemical properties of the soil samples. The bulk density was 1.39 g/cm$^3$, the content of sand, silt and clay was 38.43%, 46.38% and 15.19%, respectively, and the soil pH value was 7.0~7.5. This type of soil has large porosity, strong infiltration capacity, a shallow soil layer and low underlying permeability.

### 2.2. Experiment Design

The experiment was conducted in a completely randomized block design with 3 rainfall intensities of 60, 90 and 120 mm/h, and 7 types of soil surface microrelief measures

(Figure 1): (1) The control check, CK. It was set as flat slope surface without any surface profile undulation; (2) Alternate mounds, AM, conical, the diameter of the bulge bottom was 20 cm and the distance between each mound was also 20 cm; (3) Continuous mounds, CM, conical, the diameter of the bulge bottom was 20 cm and the distance between each mound was zero; (4) Alternate depressions, AD, inverted cone, the bottom diameter of the depression was 20 cm and the distance between each depression was also 20 cm; (5) Continuous depressions, CD, inverted cone, the diameter of the bulge bottom was 20 cm and the distance between each depression was zero; (6) Alternate mounds and depressions, AMD, the bottom diameter of the convex/concave cone was 20 cm and the distance between each mound/depression was also 20 cm; (7) Continuous mounds and depressions, CMD, the bottom diameter of the convex/concave cone was 20 cm and the convex or concave cones were set up with no distance between each other. Among these topographic relief treatments, there was only one type of surface relief pattern on the slope surface of AM, CM, AD and CD, so these four treatments were collectively named monomorphic surface relief, MSR. For AMD and CMD, both mounds and depressions were existing on the soil surface at the same time, so these two treatments were collectively named Compound Surface Relief, CSR.

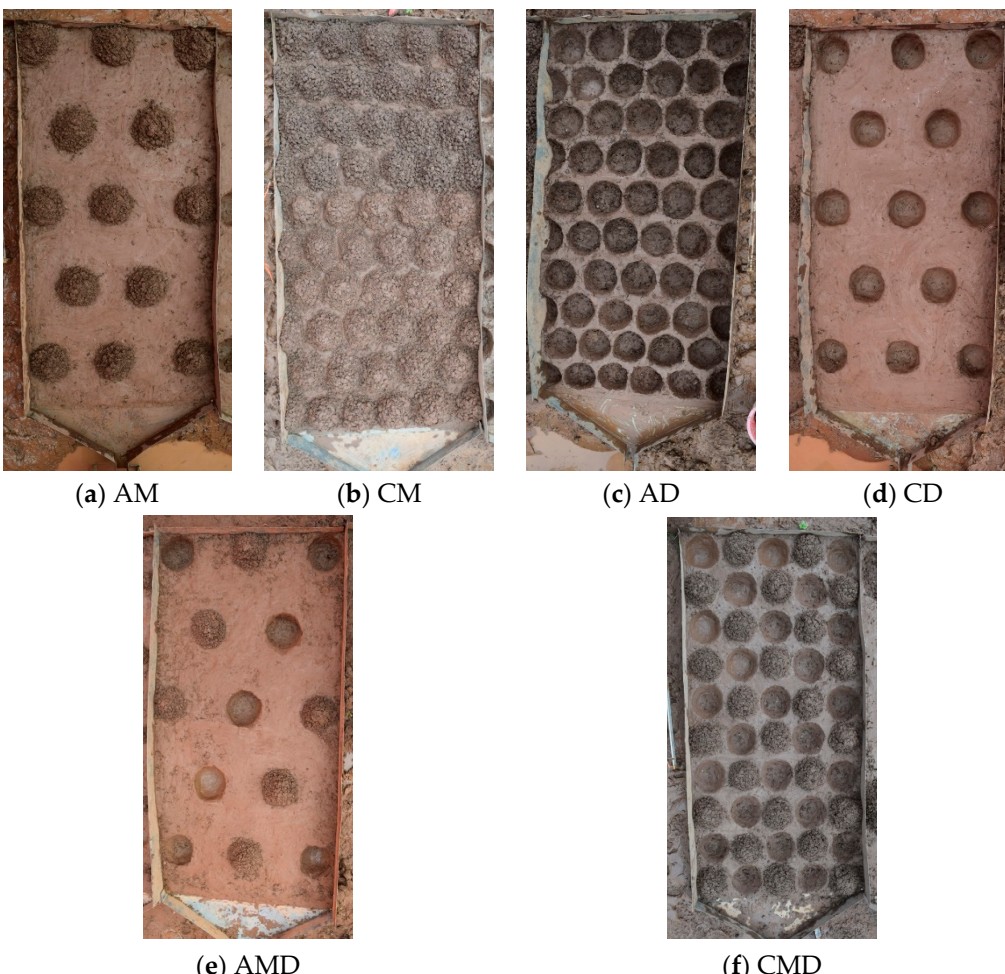

**Figure 1.** Soil surface microrelief measures.

The mounds and depressions in the experiment were handmade. A fixed-size depression was dug into the slope and the resulting soil was turned into a mound of the same size, each of the same diameter and height. The surface vegetation was removed and abandoned for 1 year in the experimental area, and 18 test plots with the size of 2 m × 1 m (length × width) were set up. The boundary of the runoff plot was sealed with an iron

sheet, which was 20 cm higher than the ground and 20 cm deep into the soil, to reduce the influence of lateral infiltration [26].

The artificial rainfall device is a portable, side-spraying field simulator with two sprinklers. The effective rainfall area of one sprinkler is 10 m$^2$, and rainfall uniformity can reach 90% when two sprinklers spray water face to face. The rainfall height was 6 m. Each treatment was repeated 3 times and was exposed to rainfall events for 90 min. The runoff generation time was recorded, and the runoff sample was collected every 2 min with a bucket.

*2.3. Runoff Parameter Measurement*

The runoff analysis included runoff generation time, runoff yield, runoff coefficient and hydrodynamic parameters.

### 2.3.1. Runoff Generation Time

Runoff is not generated as soon as the raindrops hit the ground, there is always a delay time because of the initial rainfall loss. When the continuous thin layer flow and obvious head fall appear on the soil surface, the runoff generation can be considered as the beginning. The length of the period was runoff generation time.

### 2.3.2. Runoff Yield

After runoff generated, the runoff was collected by bucket every 2 min and weighed as $M_1$. Then, the runoff samples were left standing for 24 h after weighing. After the sediment had settled, the upper layer of clear water was poured out, the water and sediment mixture samples were dried to a constant weight, recorded as $M_2$. The runoff volume every 2 min was calculated by $M_1$ minus $M_2$.

### 2.3.3. Runoff Coefficient (C)

Runoff coefficient (C) is defined as the ratio of the volume of water superficially drained during rainfall to the total volume of precipitation during a certain period. It can reflect the influence of soil surface geomorphic properties on the overland flow. It is calculated by the equation below [27]:

$$C = \frac{V_r}{V_p} \tag{1}$$

where C represents the runoff coefficient, $V_r$ is the runoff volume and $V_p$ is defined as the precipitation volume.

### 2.3.4. Hydrodynamic Parameters

The Reynolds number (Re) is the criterion of whether runoff flow is laminar or turbulent, and it can discriminate the flow pattern of runoff. When Re < 500, flow is generally laminar, and when Re > 500, flow is usually turbulent.

$$Re = \frac{Vh}{\nu} \tag{2}$$

$$\nu = \frac{0.01775}{1 + 0.0337t + 0.000221t^2} \tag{3}$$

where V is the average velocity, m/s, h is the average water depth, m, $\nu$ is the kinematic viscosity coefficient, m$^2$/s, and t is the water temperature, °C.

The Froude number (Fr) is a dimensionless number reflecting the influence of gravity on fluid motion. When Fr < 1, the runoff is slow flow; when Fr > 1, the runoff is rapid; when Fr = 1, the runoff is called critical flow. It is generally expressed as

$$\text{Fr} = \frac{\text{V}}{\sqrt{\text{gh}}} \tag{4}$$

where g is the gravitational acceleration, with the value of 9.8 m/s$^2$.

Runoff shear stress ($\tau$) is the main driving force that causes soil particle separation and sediment transport. As the runoff shear force increases, it's influence on the soil would be more effective, the quantity of stripped soil would be more and the erosion would be more serious. The equation to calculate this is as follows:

$$\tau = \rho\text{ghJ} \tag{5}$$

where $\tau$ is the runoff shear stress, Pa, $\rho$ is the density of water, kg/m$^3$, J is the hydraulic slope, approximately the sine of the slope.

### 2.3.5. Data Processing and Statistical Methods

The raw data of runoff and sediment yield was standardized, then the runoff generation time, runoff yield and runoff process were calculated by Excel as well as the graph drawing.

Both the variance analyses of the Reynolds number, Froude number and runoff shear force and the mean values of them were calculated by SPSS26.0.

## 3. Results

### *3.1. The Effects of MSR on Runoff*

### 3.1.1. The Effects of MSR on Runoff Generation Time

At the very start, the rainfall on bare soils was mainly consumed by infiltration and filling depressions. As rainfall continued, the infiltration and filling of the slope surface gradually tended to saturation, and then the runoff generation began. The time of runoff generated is the response of slope surface comprehensive effect, which is related to the surface relief, initial soil moisture content and rainfall intensity. The difference of runoff generation time can directly reflect surface fluctuation conditions when under the same initial soil moisture content and rainfall intensity conditions.

As shown in Figure 2, the average runoff generation time under 60, 90 and 120 mm/h rainfall intensity are quite different. Obviously, when the rainfall intensity is the same, the longest runoff generation time is CD, the shortest runoff generation time is AM, and it follows the rule of CD > AD > CK > CM > AM.

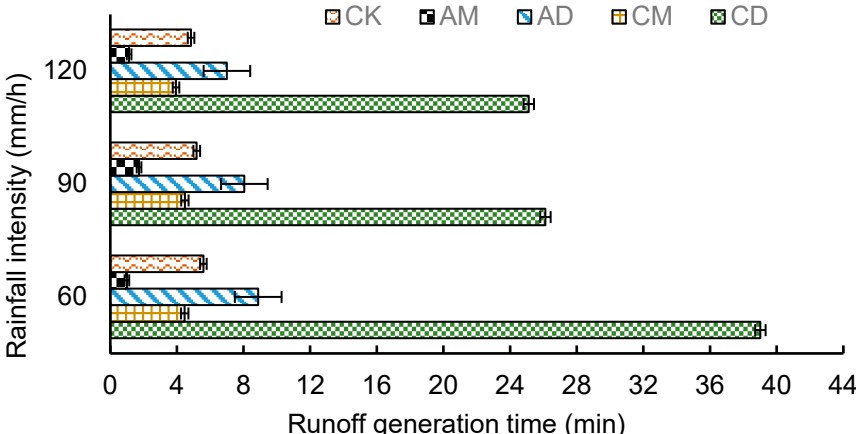

**Figure 2.** Runoff generation time of MSR treatments. MSR represents monomorphic surface relief.

For monomorphic depression treatments, depressions of CD were distributed continuously along the slope surface. Because of the collection and accumulation effects of the depression on water, runoff was only triggered once all the depressions of the soil

surface were filled. Meanwhile, the water accumulation in the depressions increased the osmotic pressure, thereby increasing the infiltration volume, delaying the runoff generation and greatly increasing the infiltration time. Therefore, the runoff generation of CD was the slowest.

Compared to CD, there were flat slope surfaces intermittent among the depressions of AD. The accumulation and interception of water, and the effect of increasing infiltration was affected by the existence of intermittent flat slope surfaces, so the runoff generation time of AD was less than that of CD. Specifically, the runoff generation time of AD was 73% shorter than that of CD.

As the surface of CK was smooth with no surface relief, the runoff generation was not affected by any terrain. In the case of natural flow generation, it neither accelerated the flow generation due to mounds, nor delayed the flow generation due to depressions. The runoff generation time was in the middle among the five topographic relief treatments.

For AM, distribution of mounds was discontinuous, and there were flat plains among the mounds. Therefore, after rainwater was rapidly divided by the mounds, the stream flow easily gathered and collected at the intermittent area and flowed downward rapidly. At the same time, due to the intermittent flat plains around the mounds, the stream easily bypassed the mounds and continued to develop into a thin layer flow when encountering the bulge barrier; therefore, it could reach the runoff collecting nozzle quickly, and so the runoff generation time was the shortest. Similarly, with the increase of bulge size, the effect of the AM group on runoff generation also increased.

### 3.1.2. Runoff Yield

Table 1 shows the total runoff yield and runoff coefficient of each topographic relief treatment in MSR. Under the condition of the same rainfall intensity, total runoff yield and runoff coefficient showed the rule of AM > CK > AD > CM > CD. In particular, the runoff coefficient of CD was far less than that of the other four treatments.

**Table 1.** The total runoff yield and runoff coefficient of MSR treatment.

| Rainfall Intensity/mm·h$^{-1}$ | Runoff Yield/m$^3$ | | | | | Runoff Coefficient | | | | |
|---|---|---|---|---|---|---|---|---|---|---|
| | CK | CM | AM | CD | AD | CK | CM | AM | CD | AD |
| 60 | 0.103 | 0.044 | 0.106 | 0.021 | 0.072 | 0.029 | 0.012 | 0.030 | 0.006 | 0.020 |
| 90 | 0.190 | 0.151 | 0.208 | 0.090 | 0.120 | 0.035 | 0.028 | 0.039 | 0.017 | 0.022 |
| 120 | 0.264 | 0.185 | 0.302 | 0.126 | 0.185 | 0.037 | 0.026 | 0.042 | 0.017 | 0.026 |

Note: MSR represents monomorphic surface relief, CK represents control check, AM represents alternate mounds, CM represents continuous mounds, AD represents alternate depressions and CD represents continuous depressions.

Table 1 also shows that the runoff coefficient of CD was the smallest out of all the three rainfall intensities. When the rainfall reached 60 mm/h, the runoff coefficient of CK, CM, AM and AD were 4.83, 2.0, 5.0 and 3.33 times of CD, respectively. Moreover, the runoff coefficient of CK, CM, AM and AD decreased to 2.06, 1.65, 2.29 and 1.29 times of CD for 90 mm/h and 2.18, 1.53, 2.47 and 1.53 times for 120 mm/h rainfall intensity, respectively. Obviously, the topographic relief could affect the function of precipitation converted into runoff, and the runoff yield and runoff coefficient were the biggest among all the MSR treatments.

There were both mounds and flat plains on the surface of AM, so the watershed effect of convex on rainfall and the storage effect of the mound's slope surface on raindrops would produce streamflow rapidly; the flat interval between the mounds made the stream flow develop freely and rapidly, thus runoff velocity of the AM surface increased compared to that of CK. With the increase of flow velocity, the infiltration decreased, so the runoff yield and runoff coefficient of AM were the biggest of all the treatments.

Because of the flat terrain, no obstruction and no retaining of surface water, the runoff loss of CK was less than that of AM. Furthermore, overflow developed naturally on mounds

to accelerate the runoff accumulation, so runoff yield and runoff coefficient of CK were second to those of AM. The runoff yield and runoff coefficient of CK were 8% and 7% less than those of AM, respectively.

Compared to AM, although the surface of AD also had flat plain intervals, there were only depressions instead of mounds existing. The evenly distributed depressions on the slope surface intercepted the runoff and accumulated the overland flow in the pits, and the runoff was not generated until the filling amount reached saturation. At the same time, the accumulated precipitation in the depressions also caused the increase of infiltration amount and the extension of infiltration time. In consequence, the initial rainfall loss became more, and the runoff yield and runoff coefficient were less than AM.

For CM, without any flat plain interval, the dense and continuous pure mounds led the stream flow around and detoured at the foot of the mounds, which significantly increased the length of the runoff path. In addition, the blocking effect of the mound on runoff was continuous, and the flow velocity of overland flow decreased, the infiltration increased and the runoff developed very slowly. Therefore, the effect of CM on runoff coefficient showed obvious enhancement compared to AM. The runoff yield and runoff coefficient of CM were 39% and 41% less than those of AM, respectively.

However, the continuous depressions on the surface of CD enhanced the accumulation and interception effect on overland flow greatly. The infiltration capacity and infiltration time of CD were greater than those of AD. At the same time, due to the lack of promotion effects on runoff of surface flat plains, CD had the most significant effect on delaying runoff, increasing infiltration and reducing runoff. Therefore, the runoff yield and runoff coefficient of CD were far less than those of other treatments. The runoff yield and runoff coefficient of CD were 37% and 43% less than those of AD, respectively.

In addition, as rainfall intensity increased, the runoff generation time was shortened, the overland flow velocity was increased and the infiltration capacity and infiltration time were reduced, so the runoff yield and runoff coefficient increased as well.

### 3.1.3. Runoff Process

Figure 3 shows the runoff process of MSR treatment under each rainfall intensity. The runoff process of all topographic measures was similar, which is shown as follows: the runoff yield increased rapidly and was unstable at the initial stage of rainfall; with the continuous rainfall, the instantaneous runoff yields gradually tended to a stable value and fluctuated slightly around the value until the end of rainfall.

The greater the rainfall intensity, the larger the fluctuation range of instantaneous water production in the middle and late rainfall, and the time required for each treatment to reach the stable runoff generation stage was also longer.

Under the three different rain intensities, the runoff generation stability of each treatment followed the rule of CM > AM > AD > CD > CK. This was because the top of the convex in CM diverged the rainfall immediately, which made it easy to form the microstream. Since the microstream could not cross the top of the convex, but only could meander along the bottom of the convex, the path of the slope surface rainfall was relatively concentrated and fixed, so the whole runoff process of CM was the most stable one.

Since there were many plains among the mounds for AM, overland flow could develop freely on this flat surface for a short time just before the microstream was blocked by another mound. As a result, the velocity and direction of the runoff changed slightly and were relatively unfixed. Therefore, the stability of the runoff process in AM came second, following CM.

Compared to the mounds, the depressions did not generate runoff until the rainfall filled up all the depressions. There would be a preferential flow path once every depression was filled. The preferential flow paths were connected with each other in series to form the thin overland flow and reach the outlet of the catchment area at last. However, with the increase of rainfall duration, the microrelief would be changed by raindrop impact and runoff scouring, and this preferential flow path could be subject to change at any time.

Therefore, the runoff process of the depressions was less stable than that of the mounds. The runoff stability of AD with plain intervals was slightly better than that of CD without plain intervals.

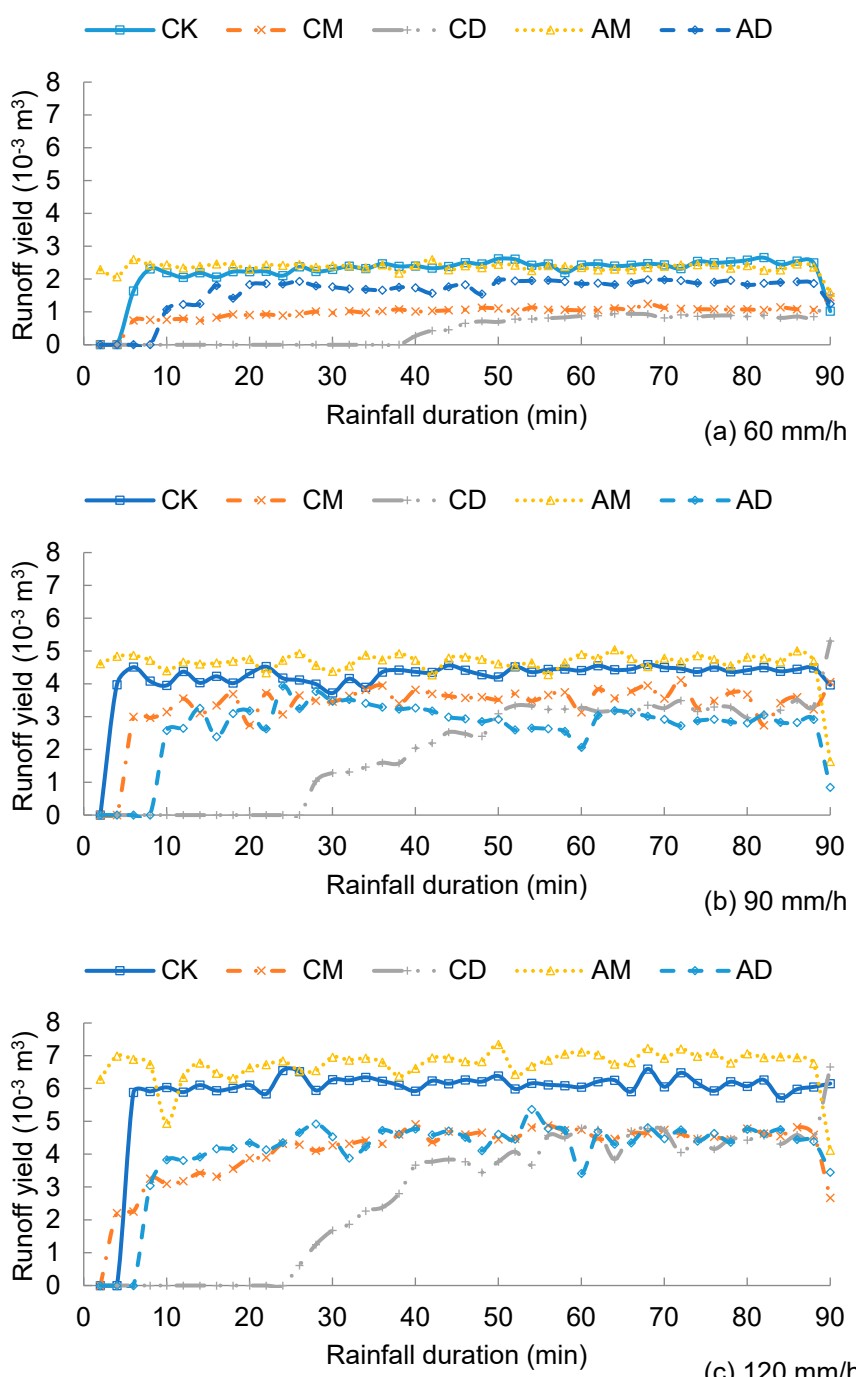

**Figure 3.** Runoff process of MSR under different rainfall intensities.

The most unstable runoff process was CK. Since the surface of CK was flat, the development of overland flow was clearly unobstructed and had a high degree of freedom. As a result, the microstream path was the most random and the runoff process was the most unstable.

Under the same rainfall intensity, the time required for runoff to reach the steady state met the following rule: CD > AD > CK > CM > AM. It was closely related to the increase of soil water content. The infiltration would reach a stable stage, and then a stable runoff

production state could be formed on the surface when the surface soil moisture content was saturated.

For AM treatment, the distributary action and raised shape of the mounds forced the runoff path to be fixed. Since this concentration showed a fixed runoff path, it was easy for topsoil to be saturated, and thus AM was the first treatment to reach the steady runoff generation state.

The continuous distribution of mounds in CM group made the runoff more obstructed and the flow velocity much lower than those in the AM group. Therefore, the runoff generation time needed to achieve a stable state for CM was slightly longer than that for AM.

Runoff was easier to form and flow on the flat soil surface. Since surface of CK was flat, the velocity of overland flow was fast, the flow time of runoff on the slope surface was short, the infiltration quantity was little and the soil moisture content increased slowly. Therefore, the time for CK to achieve the stable runoff state was longer than those of AM and CM.

Once depressions were filled, the pressure of accumulated water increased and the rainwater could infiltrate into deep soil, so the infiltration process was prolonged and the infiltration capacity was increased. Therefore, the time required to reach the saturated state was much longer, thereby prolonging the time required to achieve the runoff stable state. The continuous distribution of depressions for CD made the water storage area much larger than that of AD, so the time required to reach the runoff steady state was the longest one.

According to the research of J.A. Gomez et al. (2005), there was little difference in the runoff process between different topographic relief treatments, which is different from the above results [23]. It may be due to the design of larger slope surface in their experiment [28,29]. Compared to Zhao's research (2019) [30], the research results of this experiment are reliable.

### 3.2. The Effects of CSR on Runoff

3.2.1. The Effects of CSR on Runoff Generation Time

Figure 4 shows the comparison of the runoff generation time of CSR with that of CK. The runoff generation time was consistent with the rule of CMD > CK > AMD under the same rainfall intensity. For CMD, because of the continuous distributed mounds and depressions, the obstructed effects of mounds on runoff superimposed the intercept and storage effects of depressions on runoff, so the runoff generation time was relatively longer. However, for AMD, because of the discontinuous flat area among the mounds and depressions, the microstream formed by the mounds could be concentrated immediately, so the runoff generation time of AMD was shorter than that of CMD, and CK was in the middle. The runoff generation time of AMD was 73% shorter than that of CMD.

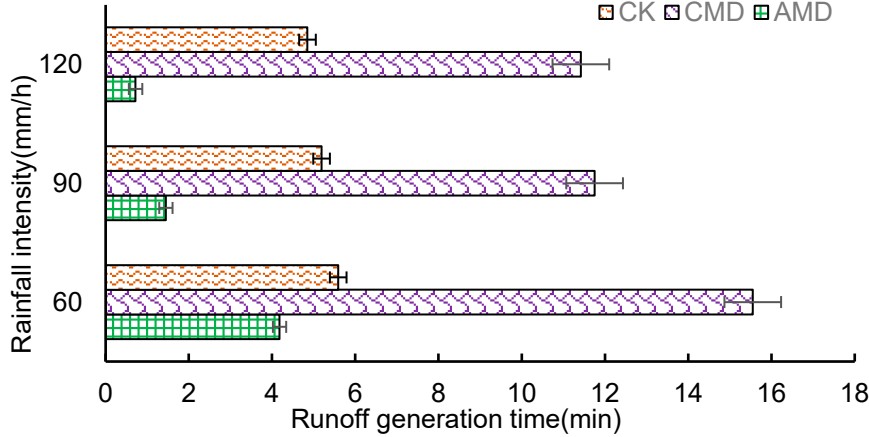

**Figure 4.** Runoff generation times of CSR treatments.

After comparing the runoff generation time of MSR and CSR, the rule of runoff generation time for each treatment under the same rainfall intensity was found easily: CM/AM < CMD/AMD < CD/AD. That is enough to prove that the exact opposite effects of mounds and depressions on runoff generation time were counteracted by CSR, so the runoff generation time presented an intermediate final state.

Summarily, the runoff generation time of CSR was between the two MSRs (CM/AM and CD/AD). Meanwhile, the runoff generation time of continuous CSR was much longer than that of alternate CSR.

### 3.2.2. Runoff Yield

Table 2 shows the total runoff yield and runoff coefficient of each CSR topographic relief treatment. Both the total runoff yield and runoff coefficient presented the trend of CK > AMD > CMD under the same rainfall intensity. For CMD, due to continuous undulation of mounds and depressions, the storage effects, blocking runoff effects and increasing infiltration effects of depressions on runoff were superimposed with the obstructing runoff effects and extending runoff paths effects of mounds on runoff. This significantly reduced the proportion of precipitation converted into runoff, so the runoff yield and runoff coefficient of CMD were low. For AMD, because mounds and depressions were distributed discontinuously, the alternate flat soil surfaces would weaken the interception effect of continuous undulating microrelief on runoff. Therefore, the runoff yield and runoff coefficient of AMD were higher than that of CMD. The runoff yield and runoff coefficient of CMD were 23% and 31% less than that of AMD, respectively.

**Table 2.** The total runoff yields and runoff coefficients of CSR treatments.

| Rainfall Intensity/mm·h$^{-1}$ | Runoff Yield/m$^3$ | | | Runoff Coefficient | | |
|---|---|---|---|---|---|---|
| | CK | AMD | CMD | CK | AMD | CMD |
| 60 | 0.103 | 0.076 | 0.038 | 0.029 | 0.021 | 0.011 |
| 90 | 0.190 | 0.123 | 0.116 | 0.035 | 0.023 | 0.016 |
| 120 | 0.264 | 0.202 | 0.174 | 0.037 | 0.028 | 0.024 |

Note: CSR represents compound surface relief, AMD represents alternate mounds and depressions and CMD represents continuous mounds and depressions.

This indicated that the runoff yield and runoff coefficient of CSR were affirmatively less than those of pure mounds topography, no matter if it was CM or AM. The runoff yield and runoff coefficient of AMD were 34% and 35% less than that of AM, respectively, and for CMD were 14% and 20% less than that of CM, respectively. However, the runoff yield and runoff coefficient of CSR were not necessarily greater than those of pure depressions topography. In other words, in terms of runoff situation, CSR decreased the runoff yield and runoff coefficient of pure mounds topography significantly, but it was not a simple neutralizing effect between promoting runoff by mounds and obstructing runoff by depressions.

To sum up, since the effect of mounds on obstructing runoff yield and depressions on intercepting and storage runoff were superimposed, the runoff yield and runoff coefficient of CSR were basically less than those of MSR.

### 3.2.3. Runoff Process

Figure 5 shows the runoff process of CSR treatment under each rainfall intensity.

For CMD, under the same rainfall intensity, runoff yield increased slowly and continuously as rainfall duration increased. In addition, the runoff process would reach a steady state after a certain time and lasted until the end of rainfall.

Continuous fluctuation in soil surface superimposed the obstruction and delay effects of mounds on runoff and the interception and storage effects of depressions on runoff, thus increasing infiltration and weakening runoff significantly. After the depressions were filled with rainwater, the runoff would be affected by raindrops hitting the water surface.

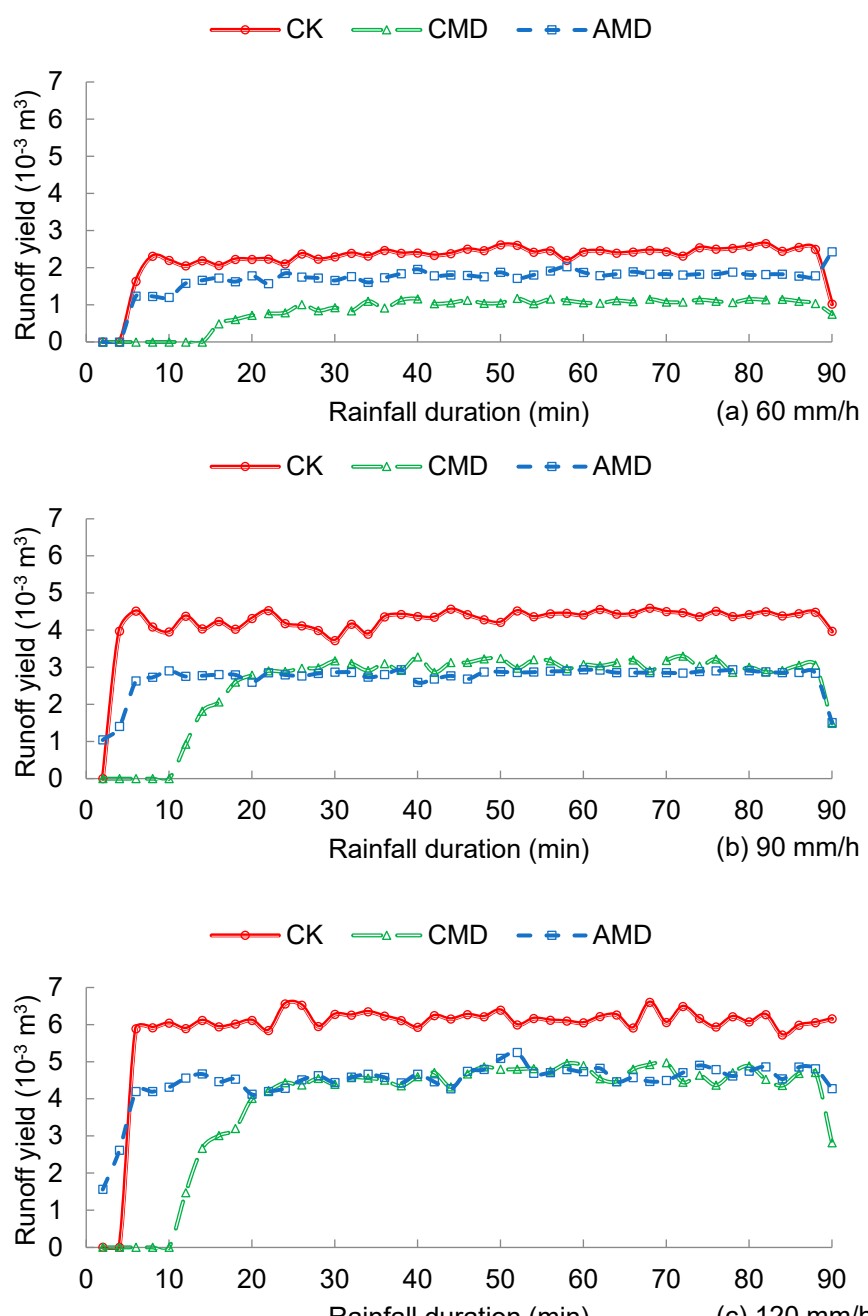

**Figure 5.** Runoff process of CSR under different rainfall intensities.

*3.3. Hydrodynamic Characteristics of Runoff*

3.3.1. Reynolds Number

Table 3 shows the Reynolds number (Re) of each treatment. It can be seen that as the rainfall intensity increased, the Re increased. In general, the Re of each rainfall event was relatively small, with an average range of 10~87 and not more than 500. According to the classical Reynolds discrimination for open channels, the runoff pattern of overland flow under simulated rainfall tests should be a "laminar flow region".

In fact, when the overland flowing water layer was shallow enough to a certain extent, it was impossible for the overland flow to keep the water surface relatively stable. The water surface tended to be unstable, and the flow transport also changed from the original consecutive to intermittent. The overland flow moved in a snowball state, which is called roll waves. The snowball movement state of overland flow could be observed during the simulated rainfall events, and this kind of overland flow had a certain capacity of

suspended sediment transportation, which made the runoff pattern different from the "laminar flow region" in the sense of open channel flow.

**Table 3.** Runoff Reynolds number of each treatment.

| Rainfall Intensity/mm·h$^{-1}$ | Reynolds Number | | | | | | |
|---|---|---|---|---|---|---|---|
| | CK | CM | CD | AM | AD | CMD | AMD |
| 60 | 27 | 24 | 20 | 23 | 18 | 10 | 17 |
| 90 | 47 | 37 | 29 | 42 | 32 | 34 | 28 |
| 120 | 87 | 31 | 28 | 75 | 48 | 44 | 45 |

The resistance coefficient increased as the energy consumption of runoff increased. Through reducing the resistance coefficient of runoff, overland flow minimized the energy consumption in order to promote the flow to form a new state of equilibrium. The snowball flow was another equilibrium state of thin layer flow, which made the overland flow smoother by sacrificing some fluid particles.

### 3.3.2. Froude Number

Table 4 showed the Froude number (Fr) of the thin layer flow on the slope surface for each treatment under different rainfall intensities.

**Table 4.** Froude number of each treatment.

| Rainfall Intensity/mm·h$^{-1}$ | Froude Number | | | | | | |
|---|---|---|---|---|---|---|---|
| | CK | CM | CD | AM | AD | CMD | AMD |
| 60 | 0.057 | 0.027 | 0.044 | 0.046 | 0.061 | 0.058 | 0.057 |
| 90 | 0.071 | 0.030 | 0.047 | 0.041 | 0.042 | 0.066 | 0.060 |
| 120 | 0.059 | 0.044 | 0.047 | 0.036 | 0.040 | 0.060 | 0.047 |

Compared to CK, the Fr of CSR and MSR decreased about 40.17% and 64.53% under 60 mm/h rainfall intensity, respectively. The Fr of CSR and MSR decreased about 58.43% and 59.22% under 90 mm/h rainfall intensity, respectively. Under 120 mm/h rainfall intensity, the Fr of CSR and MSR decreased about 65.14% and 67.43%, respectively. This indicated that, with rainfall intensity increased, the runoff pattern of CSR and MSR were more similar to each other and the retarding effects of topography on overland flow were more effective.

### 3.3.3. Runoff Shear Stress

Table 5 showed the runoff shear stress of each treatment. The runoff shear stress of both CSR and MSR were higher than those of CK under all three rainfall intensities. Compared to CK, the runoff shear stress of MSR and CSR increased about 17.86~103.7% under 60 mm/h rainfall intensity, about 32.69~115.7% under 90 mm/h rainfall intensity, and about 26.84~70.27% under 120 mm/h rainfall intensity. It demonstrated that for CSR and MSR, with rainfall intensity increased, the patterns of overland flow were more similar and the reducing effects on runoff were more efficient.

**Table 5.** Runoff shear stress of each treatment.

| Rainfall Intensity/mm·h$^{-1}$ | Runoff Shear Stress | | | | | | |
|---|---|---|---|---|---|---|---|
| | CK | CM | CD | AM | AD | CMD | AMD |
| 60 | 2.805 | 5.714 | 4.125 | 3.970 | 3.306 | 3.511 | 3.527 |
| 90 | 2.420 | 5.219 | 3.861 | 4.223 | 4.160 | 3.211 | 3.401 |
| 120 | 2.731 | 4.066 | 3.878 | 4.650 | 4.318 | 3.464 | 4.018 |

## 4. Discussion

Different topographic relief treatments showed different physical morphological characteristics of depressions and mounds, resulting in different soil infiltration rates and runoff directions, then affecting the process and results of runoff generation on slope surface.

Generally, the depressions would intercept the net rain, increase infiltration, block runoff to some extent and delay runoff generation and catchment processes. Therefore, for the depressional slope surface, the runoff was generated only after all the depressions were filled with rainwater. This phenomenon was mentioned before by Darboux (2005) and Thompson et al. (2010) [31,32]. Continuously distributed depressions would accumulate the rainwater, increase osmotic pressure and increase infiltration, thereby hindering the slope flow generation. If there were flat intervals distributed among these depressions, the water storage effects would be weakened and the flow generation time would be shorter than that of continuously distributed depressions. The influence of surface mounds and depressions on runoff in microtopography is totally different. The micro slope surface and micro aspect of mounds changed the runoff process and direction. The slope flow wase tortuous for the blockage of the mounds, and then changed the path and confluence direction of runoff. For one thing, the runoff velocity and sediment carrying capacity were reduced due to the tortuosity of runoff, which blocked the runoff generation, for another, the existence of mounds increased runoff depth and runoff shear stress, thus promoting the generation of runoff. This is similar to the research of Zhao (2015) and Fox et al. (1998) [33,34]. In addition, whether there were flat intervals or not changed the flow generation mechanism as well.

When the mound distribution was discontinuous, the rainwater was quickly diverted by the water diversion effect of the convex, and the small tributaries were gathered into a stream easily at the flat interval and then flowed downhill quickly. At the same time, the flat intervals made the micro streams bypass the bulge and continue to develop into overland flow much easier when they encountered the barrier of the mounds, which made overland flow reach the downstream quickly with the shortest flow generation time. Meanwhile, when the mounds were distributed continuously, it was hard for the micro streams to be gathered because of the barrier of continuous mounds. The runoff was blocked by bumps everywhere in the flow process, with a tortuous flow path and relatively slow flow rate, so the flow generation time was longer than that of the intervals of mounds.

Mounds and depressions on soil surface have completely different effects on runoff generation and accumulation. Generally, mounds promote runoff generation while depressions delay runoff generation, but the distribution of mounds and depressions makes their effect on runoff become much more complicated: (1) When the mounds or depressions exist separately, the continuously distributed mounds or depressions can hinder the accumulation and development of runoff. The influence of interspaced depressions on runoff is weaker than that of continuous depressions, while interspaced mounds can promote runoff development. (2) When the mounds and depressions exist simultaneously, the watershed effect of mounds on promoting runoff production are weakened, while the blocking effect of the mounds' geometry on runoff obstruction is strengthened, which combines with the interception and accumulation effect of depressions on runoff, consequently forming a more significant effect of blocking the development of runoff. (3) The flat interval buffered the promoting effects of mounds or inhibiting effects of depressions on runoff generation.

The undulation of microtopography affects the generation and confluence of runoff on the slope by affecting the runoff generation time, infiltration [35], runoff yield and runoff process, and then affecting the soil moisture and erosion status. Therefore, by separating the two morphological elements of the mounds and depressions that make up the microtopography to study their impact on the runoff, the internal impact mechanism of the mounds and depressions on the slope runoff can be clearly explained, as well as the interaction quantified, which lays a foundation for the later study of the influence of the mounds and depressions on the slope hydrology connectivity, sediment connectivity and soil erosion. That is the important scientific significance of this study.

Compared to other studies based on microtopography in recent years [36–38], this study clarified the influence mechanism of the geometric shape and physical characteristics of microtopography on runoff and revealed the root cause of the many impacts of microtopography on runoff and soil moisture. Different from the composite microtopography that both mounds and depressions existed simultaneously [32,37,38], separately studying of these two patterns of microrelief can reveal the essential impact of different topographies on runoff, which is the biggest innovation of this work. By stripping the interaction between mounds and depressions, their separate action mechanisms can be studied and a comparative analysis with the interaction mechanism can be made as well. Consequently, the comprehensive effects of microtopography on runoff with and without the interaction between the mounds and depressions can be clearly explained.

By comparing the effects of CSR and MSR on runoff, it can be found that the flow reduction effects of MSR were much better than those of CSR. Therefore, in practice, a certain degree of microrelief consisting of mounds and depressions on soil surface is conducive to soil and water conservation. Instead of flat slope, people can create mounds and depressions on purpose to conserve soil and water.

However, the inadequacy of this study is that the research scale is too small. The size of microtopographic relief is at the centimeter-level, while it is often at the meter-level in natural situation. Later, the research scale should be expanded, and more soil types should be researched to do comparative research for further study on the influence of microtopographic relief patterns on the triggering mechanism of slope soil erosion.

## 5. Conclusions

Surface mounds can promote surface runoff triggering and shorten runoff generation time significantly, while surface depressions can obstruct surface runoff triggering and prolong runoff generation time. The runoff generation stability of each treatment for MSR followed the rule of CM > AM > AD > CD > CK under the same rainfall intensity and size. In addition, the time required for each treatment to reach the steady runoff state followed the rule of CD > AD > CK > CM > AM.

Whether there was an interval between mounds or depressions also affected the characteristics of runoff. The runoff-promoting effects of mounds were totally exhibited by AM, while, in CM, continuous mounds blocked the waterway, making the runoff generation time of CM 5 times as much as AM, and runoff yield and runoff coefficient of CM 39% and 41% less than those of AM, respectively. Runoff generation time of CD was 3.8 times that of AD, and runoff yield and runoff coefficient of CD were 37% and 43% less than those of AD, respectively.

CSR can significantly neutralize the flow-promoting effects of the mounds and the flow-inhibiting effects of the depressions, making the runoff yield and runoff process present a neutral state between the mounds and depressions. CSR prolonged runoff generation time from 1–10 min of MSR to 5–16 min. The runoff yield of CSR presented as 0.12, between 0.17 for mounds and 0.10 for depressions, and the runoff coefficient of CSR was also a neutral value, 0.021, which was between 0.029 for mounds and 0.018 for depressions.

The Re of this experiment was small, and the runoff pattern of different rainfall intensities under simulated rainfall belonged to a "laminar flow region". In fact, the overland flow moved in a snowball state, which is called roll waves. The runoff pattern was different from the "laminar flow region" in the sense of open channel flow. The larger the rainfall intensity was, the more similar the runoff pattern of undulating slope was and the stronger the retarding effect of runoff was. Our results demonstrated that for CSR and MSR, with rainfall intensity increased, the pattern of overland flow was more similar and the reducing effects on runoff were more efficient.

**Author Contributions:** Conceptualization, X.L. (Xinlan Liang) and Y.W.; data curation, J.F., Z.Y. and L.Z.; investigation, X.L. (Xinlan Liang) and J.F.; methodology, X.L. (Xinlan Liang), J.F. and Z.Y.; project administration, X.L. (Xinlan Liang); resources, X.L. (Xinlan Liang), L.Z., J.L. and Y.W.; software, J.F., Z.Y. and L.Z.; validation, X.L. (Xinlan Liang); visualization, X.L. (Xinlan Liang) and J.F.; writing—original draft, X.L. (Xinlan Liang) and J.F.; writing—review & editing, J.F., Z.Y., L.Z., J.L., X.L. (Xiuyuan Lu), S.Z., Q.L., Z.Z. and Y.W. All authors have read and agreed to the published version of the manuscript.

**Funding:** This work was supported by the National Natural Science Foundation of China (42277326), China Post-doctoral Science Foundation (2020M683368), Sichuan Province Science and Technology Support Program (2023NSFSC0119 and 2021YJ0544) and the Project of undergraduate Education and teaching Reform Research of Sichuan Agricultural University (201929).

**Institutional Review Board Statement:** Not applicable.

**Informed Consent Statement:** Not applicable.

**Data Availability Statement:** Data are available from the authors upon request.

**Conflicts of Interest:** The authors declare no conflict of interest.

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
