# Peer review of "The Effect of Soil Surface Mounds and Depressions on Runoff"

_sustainability, doi:10.3390/su15010175_

Round 1

Reviewer 1 Report

Introduction: A lack of this manuscript is the absence of a thorough and careful review of the literature on some topics. In particular, reading the introduction, I expected to see several experimental approach, largely used to investigate the runoff. In my opinion, the references cited for this argument are not sufficient to give solidity to your hypothesis. In particular, to improve the introduction, you could consider following recent (but not limited to) studies (published on Sustainability MDPI):

https://doi.org/10.1016/j.iswcr.2015.10.001

https://doi.org/10.3390/su13116058

https://doi.org/10.3390/su13041991

Results and discussion: In my opinion, the most severe lack of this paper. I suggest to use more literature studies (by experimental tests or field measured) to validate your experimental results. With regard to the discussion section, I would suggest to also focus on what has been presented and how this can be improved in further development of your research.

Conclusions: You have to better specify the novelty of this work.

Reviewer 2 Report

The manuscript "The effect of soil surface mounds and depressions on runoff" was submitted to Sustainability. It's a nice research about soil erosion. I have some concerns and my decision is minor revision, finally I recommend to authors that improve the quality of this nice paper as following.

1.    The research results should be stated more accurately in the abstract. This will be of great help to the readers in the future and has an effective role in attracting the readers.

2.    Please talk more about the results and conclusions in the abstract to improve the quality.

3.    There are repeated words between the title and keywords, please revise them, please be careful.

4.    Lines 87 to 93 have no references and the authors should speak in a documentary manner.

5.    How was obtained the information about line 95-97?

6.    In the materials and methods section, it is necessary to mention the statistics methods in more detail.

7.    The results are well written. congratulations

8.    One of the significant concerns is that the authors should carefully develop a discussion section to talk about the significance, shortages or advantages of the methods you proposed, the reliability and meaning of your results (compared to other related studies) etc.

9.    Please be sure that all the references cited in the manuscript are also included in the reference list and vice versa with matching spellings and dates.

10.                   Finally, I checked plagiarism detection of this research and the similarity is 12%, please checked attached file.

Reviewer 3 Report

Dear authors,

The manuscript is based on controlled laboratory study and tried to find the role of depressions and mounds on runoff. I have some queries :

1. Rainfall intensities of 60, 90 and 120 mm/hr were chosen for study. What was the criterion for choosing it? Does it represent natural rainfall of the region?

2. It was a simulated rainfall experiment but no information about the type of simulator used is given.

3. The study was conducted in 2 m X 1 m plots. There is no mention about the level of soil slope. Was it completely flat or some fixed slope was used? If slope was given what was the percent slope and why exactly that slope was used. As it is laboratory experiment, why it was not conducted using different level of slopes?

4. In the study seven types of soil surface micro-relief was used. Does these type of micro-relief represent actual field conditions. 

5. How these depressions and mounds were prepared?

6. Was the experiment done on natural undisturbed soil condition or the disturbed soil surface was prepared? 

There is need to correlate the outcomes of the study with its practical utility of soil erosion and conservation.

It would have been better if different type of soils may have been used in the study.

Regards

Round 2

Reviewer 1 Report

The article was improved following the reviewers' suggestions. Congratulations.

Reviewer 3 Report

Dear authors, almost all my queries are answered. 
